# Evolutionary Study of Protein Short Tandem Repeats in Protein Families

**DOI:** 10.3390/biom13071116

**Published:** 2023-07-13

**Authors:** Pablo Mier, Miguel A. Andrade-Navarro

**Affiliations:** Faculty of Biology, Institute of Organismic and Molecular Evolution, Johannes Gutenberg University Mainz, 55128 Mainz, Germany; andrade@uni-mainz.de

**Keywords:** low complexity regions, repeats, protein sequence analysis

## Abstract

Tandem repeats in proteins are patterns of residues repeated directly adjacent to each other. The evolution of these repeats can be assessed by using groups of homologous sequences, which can help pointing to events of unit duplication or deletion. High pressure in a protein family for variation of a given type of repeat might point to their function. Here, we propose the analysis of protein families to calculate protein short tandem repeats (pSTRs) in each protein sequence and assess their variability within the family in terms of number of units. To facilitate this analysis, we developed the pSTR tool, a method to analyze the evolution of protein short tandem repeats in a given protein family by pairwise comparisons between evolutionarily related protein sequences. We evaluated pSTR unit number variation in protein families of 12 complete metazoan proteomes. We hypothesize that families with more dynamic ensembles of repeats could reflect particular roles of these repeats in processes that require more adaptability.

## 1. Introduction

Tandem repeats in proteins are periodic sequences consisting of several identical or similar units of residues adjacent to each other [1]. They are common in protein sequences [2,3]. The length of the units ranges from one to more than 60 [4]. Even if they emerge as the repetition of identical units, they may mutate and lead to degenerated tandem repeats. In the case of repeats forming assemblies of repeated structural units, even when degenerated, they will still conserve their general structure [5].

The detection of tandem repeats relies on a direct search of repetitive elements in a sequence or structure using either homology search [6], clustering [7], with Hidden Markov Models [8], or taking advantage of the modularity of protein structures as defined in PDB entries [9,10]. Computational approaches have been developed for the detection of tandem repeats in genomic DNA (e.g., TriCoLOR [11]) and proteins (e.g., REP2 [12]). RepeatsDB is a database of tandem repeats in protein structures [13], which is also used as a library to predict tandem repeats in RepeatsDB-lite [14].

The simplest form of tandem repeats are homorepeats, consisting of units of length one. Homorepeats such as polyglutamine and polyalanine regions have been reported to grow in length due to replication slippage [15,16]. This process occurs when the DNA polymerase encounters a repeated sequence and results in the addition of a new unit to the existing repeat or in the deletion of a repeated unit during the replication process [17].

In a recent study where we assessed simple repeats in protein sequences, we noted their high frequency and prevalence across protein families in many organisms [18]. In particular, we observed very often that some protein families display bursts of short repeats, usually in disordered regions, and we observed tendencies for particular repeat lengths in a few organisms. Particular protein families can have functions that put them under evolutionary pressure for tandem repeat unit variation because this variation gives them some selective advantage. For example, we observed the pressure to accumulate polyQ in protein families with many protein interactions, and we used this as support for the function of polyQ in modulating protein interactions in vertebrates [19]. Similarly, we would expect that the characterization of the evolutionary expansion of given short repeats in particular protein families and species could reflect a functional role of such repeats. The association of regions biased for a few amino acids and functions are known in RG-rich regions in liquid–liquid phase separation and RNA binding-related proteins [20] and in RS-rich regions in splice-related plant proteins [21]. We hypothesized that a search for possible events of tandem duplication taking advantage of large numbers of sequences in protein families could serve to point to protein families and repeats under strong selective pressure for duplication, which could be helpful for their functional characterization.

Here, we present a strategy to identify variation in the number of units in protein short tandem repeats (pSRTs) in a set of homologous sequences using pairwise alignments. We provide this method as a web tool that takes as input a set of (ideally homologous) proteins and can be accessed at http://cbdm-01.zdv.uni-mainz.de/~munoz/pSTR/ (accessed on 5 July 2023). Application of this method to protein families from 12 complete metazoan proteomes revealed the variable levels of evolution of these units across different protein families.

## 2. Materials and Methods

We downloaded a set of 70 expert-curated reference groups of orthologs (RefOGs) with proteins from 12 bilaterian species from the latest update of the OrthoBench benchmark suite [22]: *Caenorhabditis elegans* (cel), *Drosophila melanogaster* (dme), *Ciona intestinalis* (cin), *Danio rerio* (dre), *Tetraodon nigroviridis* (tni), *Gallus gallus* (gga), *Canis familiaris* (cfa), *Monodelphis domestica* (mdo), *Rattus norvegicus* (rna), *Mus musculus* (mmu), *Pan troglodytes* (ptr) and *Homo sapiens* (hsa). We also downloaded the complete reference proteomes of these species from UniProtKb release 2022_05 [23]. For the calculation of the pSTRs, we did not take into account protein sequences with ‘X’ residues.

## 3. Results

### 3.1. Protein Short Tandem Repeats Search Strategy

Protein Short Tandem Repeats (pSTRs) are residue units identical in sequence and directly adjacent in a protein sequence. We compute them in a sequence using the following procedure. For all lengths from two to half of the protein length (the theoretical maximum length of a pSTR to have more than one unit), a window of said length is slid through the protein sequence; if there are two or more consecutive equal units, the unit and the number of times it is repeated are stored (Figure 1). All patterns are reported, even if they are overlapping. We did not consider pSTRs of length one because it is relatively easy to find two consecutive identical amino acids in a sequence by chance. The “tandem” in pSTR refers to repetitions of the same unit; in Figure 1, for example, there are four different pSTRs, with units “ML”, “ACD”, “PQ” and “IL”.

When in a comparison between two protein sequences a pSTR unit fully aligns to a gap (green in Figure 1C) and is adjacent to a conserved pSTR (in pink), this highlights an event of gain or loss of a pSTR unit. To detect these, we do an all-versus-all pairwise comparison using MUSCLE v3.8.1551 with default parameters [24]. Two additional aligners were tested in the development phase: ClustalO v1.2.4 [25] and MAFFT v7.453 [26]; the results obtained with them were comparable, but MUSCLE was at least three times faster than the others. Our pipeline limits possible misalignments of the repeats by focusing only on pairwise alignments, and by using homologous proteins in the input dataset, so that the alignments are simplified by a high sequence similarity.

For a direct comparison between the pSTR unit variants found within a family, they are mapped to positions in one sequence from the initial input dataset selected as query. Each protein for which we find pSTR variation is aligned with the query (with MUSCLE v3.8.1551 and default parameters [24]), and the position in the query sequence aligning with the central position of the extra unit is considered as its mapped position.

The query sequence itself may not share any residue with the extra unit since these are identified in alignments between different proteins within the input dataset, but the mapping to the query reflects the position of the repeat variant in the family (considering that the dataset comprises proteins from one protein family). In this way, it is possible to identify hotspots of repeat evolution of the entire family using the query sequence as reference.

### 3.2. Comparison of pSTRs in Metazoans

We identified the pSTRs in 12 complete metazoan proteomes. These are relatively frequent, with an average of 3.2% residues of the proteomes being in pSTRs, with frequency values ranging from 2.5% in *C. intestinalis* to 4.4% in *D. melanogaster* (Figure 2a). For this calculation, we took into account the proportion of residues in the proteomes participating in at least one pSTR, although we acknowledge that short pSTRs with units composed of very prevalent amino acids may be occurring randomly.

To add details to the differences in amino acid composition, we then calculated the amino acid frequencies in pSTRs comparing them to their individual amino acid background usage (Figure 2b). Amino acids with a low prevalence in proteomes tend to be even rarer in pSTRs and that is the case of cysteine, phenylalanine, histidine, methionine, tryptophan and tyrosine. The use of frequent amino acids in pSTRs is variable. Some amino acids are less used: leucine, isoleucine, asparagine, threonine, valine; some are more used: alanine, glutamic, glycine, proline, serine; and some are equally used: aspartic, lysine, glutamine, arginine.

Regarding species-specific trends, while the backgrounds are relatively constant, there is variation in the amino acid composition of pSTRs. Threonine and asparagine are more abundant in the pSTRs of *C. elegans*, *D. melanogaster* and *C. intestinalis*, and glutamine stands out in *D. melanogaster* (as already described [27]), to the best of our knowledge yet without a functional explanation. These results confirm and extend trends of repeat composition biases associated with particular species and suggest that different evolutionary pressures, that could be related to particular species and suggest that different evolutionary pressures, and that could be related to function, might influence the types and compositions of pSTRs.

### 3.3. Analysis of pSTR Unit Variation

Next, we downloaded the set of 70 Reference Orthologous Groups (RefOGs, or COGs) with proteins from the same 12 species as in the previous section, provided by OrthoBench (see Materials and Methods), the standard benchmark to assess the accuracy of orthogroup inference methods. For each COG, we calculated pSTR unit variation. For a total of 24,345 pSTRs, we identified 81 events of unit variation (Appendix A). Only one COG had no pSTRs, indicating that while pSTRs take roughly 3% of proteomes, they are widely distributed across families. Differently, events of pSTR unit variation were identified in only 22 COGs (31%). We note that this number depends on the choice of species and that more events of pSTR unit variation might be found if more species are investigated. Seven of the COGs had more than five events, suggesting that several types of repeats often evolve together in the same family.

Since low complexity regions residing within disordered regions were a feature previously identified as enriched in proteins involved in phase separation [28], here we decided to compare our results in COGs with the property of proteins of being known or predicted to phase separate. A small dataset of 89 human Liquid–Liquid Phase Separation (LLPS) drivers [29] had an overlap of only one protein to our 99 human proteins in families with pSTR unit variation, not allowing a meaningful statistical analysis. We then used the FuzDrop method [30], which scores all human proteins based on their propensity to drive LLPS from zero (low) to one (high). With a cut-off of 0.75, 6239 out of 20,366 human proteins are selected, and 40 of our 99 proteins in families with pSTR unit variation (*p*-value = 0.024). The significance increases when using a stricter cut-off of 0.90, which selects 4707 out of 20,366 human proteins, 35 out of 99 in the families with variable pSTRs (*p*-value = 0.004). We conclude that, at least according to predictions, pSTR variation is associated with involvement in LLPS.

To illustrate an event of repeat unit variation in a family, we use the E3 ubiquitin ligase PQT3-like protein (PQT3L) from *Arabidopsis thaliana*, which we used previously [18] to show a family with repeat variation. Here, we manually tuned the selection of homologs to illustrate individual events of variation for two groups of tandem repeats close to each other and included the paralog PQT3 (Figure 3).

The multiple sequence alignment of the *A. thaliana* PQT3L protein with homologs indicates the variation of repeats of ‘GP’ (one to four units), and close in sequence, the variation of seven amino acid repeats with sequence ‘QPGFNGV’ or similar (one to five units) (Figure 3A). There is no correlation between the number of units of each of the ensembles. The phylogenetic tree of these sequences can be used to infer consecutive and independent events of unit expansion (Figure 3B). We show the AlphaFold model of the *A. thaliana* protein structure just to illustrate that it leaves large regions not predicted, suggesting that they could be disordered (Figure 3C). This example suggests that PQT3L has a higher pressure for repeat unit variation than PQT3.

In the example, the mapping of unit variations obtained for the 10 sequences shown in the figure occupies six positions when mapped to the *A. thaliana* PQTL3: 508 and 509 for ‘GP’, 518 and 529 for ‘QPGFNGV’ and the similar ‘QPGFNGF’, and then there is unit ‘NNN’ at positions 651 and 654. In the MobiDB database of protein disorder [31], most of this protein is predicted to be disordered (74%) with the consensus of several disorder predicting methods. Considering that the disordered regions of *A. thaliana* PQTL3 are predicted to be in amino acid positions 78–207, 236–241, 270–288 and 358–892, it is intriguing to note that most of the variable units occur in the small region shown in the alignment (508–529). Examples of protein-driven phase separation are known to occur in *A. thaliana* [32,33], and indeed E3 ubiquitin ligases are known to regulate phase separation and form part of phase-separated organelles [34]. Thus, it could be possible that this E3 plant family uses and regulates LLPS similarly.

### 3.4. Web Tool to Search for pSTR Unit Variation

We developed a web tool called pSTR (http://cbdm-01.zdv.uni-mainz.de/~munoz/pSTR/; accessed on 5 July 2023) that only needs as input a set of protein sequences in FASTA format. The tool identifies pSTR unit variation following the steps described above. The results are provided both raw (list of protein pairs with at least one event of pSTR unit variation, and their mapped position in the query) and processed. The latter includes a heatmap depicting the protein pairs that support at least one event of repeat unit variation (sequences in the same order as in the input file). In addition, the tool provides a representation of the positions of all variations mapped in the query.

Execution time depends on the number of sequences in the input file and on their length. As a guideline, a search with 100 sequences as input with a mean length of 226 amino acids executes in less than a minute.

As example for the use of the pSTR web tool, we provide the dataset containing the 10 homologs of PQT3/PQT3L used in Figure 3 (Appendix A, and available in the web tool as an example dataset). The execution takes approximately 7 s.

Furthermore, we offer the user the option to run their analysis with the standalone version of the code. In it, there is the additional possibility to start the execution with just one protein in FASTA format. In that case, the program searches for homologs of the input protein (which is considered as the query sequence for the downstream mapping of the pSTR unit variants) in a set of 100 mammalian proteomes (Appendix A), one protein per proteome (proteomes downloaded from the UniProtKB database release 2022_05 [23]). The search is performed with BLAST v2.9.0+ and default parameters [35]. The hypothetical selection of protein fragments as homologs would not compromise the results of the execution, but rather would minimize the number of pSTR with unit number variation obtained. Then, the search for pSTR starts with the generated multifasta file of all homologous sequences (or with the provided multifasta protein dataset).

## 4. Conclusions

We have developed a method to detect protein Short Tandem Repeats (pSTR) in protein sequences and to assess their unit variation with the aid of homologous sequences. Our analysis of a set of 70 curated groups of orthologs allowed us to report the properties of pSTRs and their variation. pSTRs cover 3% of sequences. While pSTRs are widely distributed across protein families (only one had no pSTR), the 81 events of repeat unit variation we found occurred in just 22 groups, suggesting that these events have a tendency to cluster. We found a significant association of these events to families of proteins predicted to be involved in LLPS, which suggests that evolutionary pressure for repeat unit variation could come from an adaptive function related to protein phase formation.

Considering that our analysis of pSTR unit variation was based on a fixed set of 12 very taxonomically distant metazoan species, it is remarkable that we detected many events. Using these species was necessary for an automated procedure that allowed us to extract general conclusions about pSTR composition and unit variation at the level of Metazoa. A finer analysis is illustrated with our example of *A. thaliana* PQT3/PQT3L proteins. Manual exploration and iterative selection of homologs allowed to produce a refined set of plant proteins (more specifically, belonging to the malvids taxonomic clade), in which it is relatively easy to track individual events of pSTR unit duplication and mutation.

In any case, we understand that further work is needed to evaluate the association of particular types of pSTRs and mechanisms for their evolution with different taxa. In this first approach, we wanted to focus on the set of species featured in OrthoBench, to demonstrate the relevance of pSTR evolution as a driving force for variation in disordered regions of particular protein families.

To aid the exploration of particular families and the iterative selection of homologs, we have implemented our method as a web tool. With the continuous increase in the number of species and proteomes sequenced, we foresee that tracking the evolution of pSTRs will become increasingly simple. This will help the characterization of regions with repeats and low composition bias, revealing protein families and domains under evolutive pressure to change the number of repeat units, allowing us to understand how these regions evolve and to detect functions associated with them.

## Figures and Tables

**Figure 1 biomolecules-13-01116-f001:**
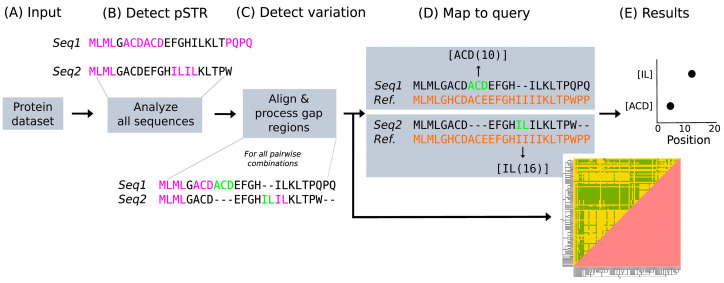
Workflow for the identification of pSTR unit number variation in protein datasets. (**A**) To identify pSTR and their unit number variation, we need a protein dataset in FASTA format. (**B**) Sequences are analyzed independently to search for pSTRs, in pink. (**C**) Then, sequences are aligned pairwise and unit variations are detected (in green) if they align to a gap and are repeated either N- or C-terminally. (**D**) Each sequence is then aligned with the query, which must be one of the proteins in the input dataset, to positionally map the pSTR with unit number variation and annotate them as [UNIT(mapped position)]. (**E**) Results are presented per pSTR with unit number variation and in a heatmap showing the protein pairs for which there is at least one of them.

**Figure 2 biomolecules-13-01116-f002:**
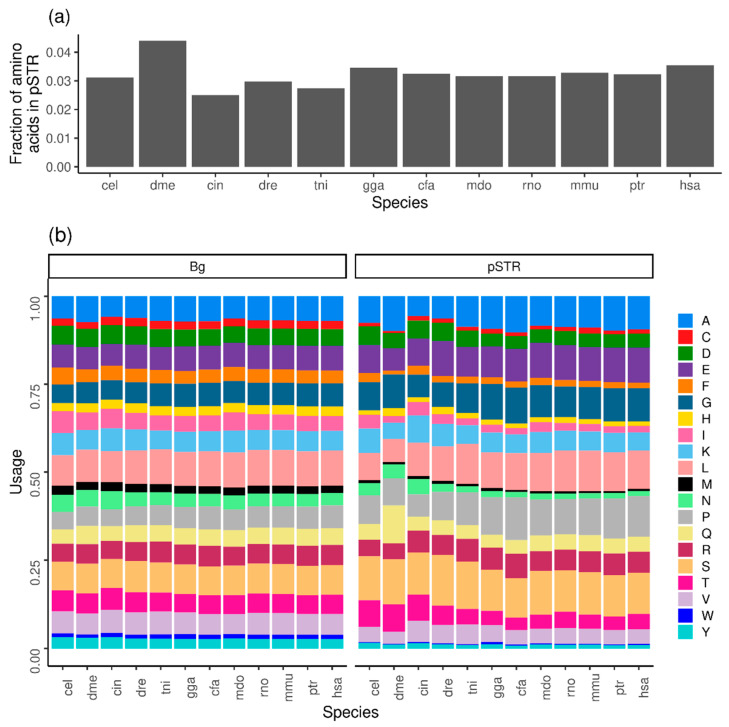
Properties of pSTRs in 12 species. (**a**) Fraction of amino acids in pSTRs. (**b**) Amino acid usage in background (whole proteome, **left**) and pSTRs (**right**). Species are annotated by a three-letter code (see Section 2).

**Figure 3 biomolecules-13-01116-f003:**
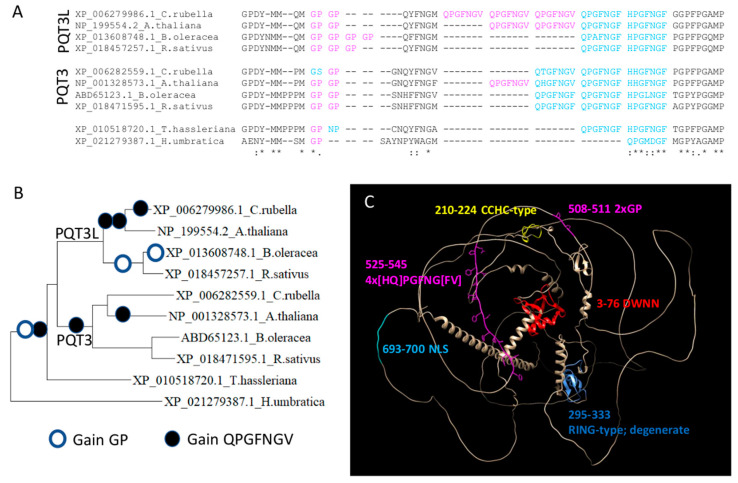
Variation of unit number of two groups of tandem repeats in plant E3 ubiquitin ligase PQT3 and PTQ3L proteins. (**A**) Part of the multiple sequence alignment of *Arabidopsis thaliana* E3 ubiquitin ligase PQT3 (UniProtKB:F4JP52) and PQT3-like (UniProtKB:B9DFV2) proteins with homologs from *Capsella rubella*, *Raphanus sativus*, *Brassica oleracea*, *Tarenaya hassleriana* and *Herrania umbratical*. Perfect repeats (‘GP’ or ‘QPGFNGV’) or close variants are highlighted in pink and blue, respectively. (**B**) Phylogenetic tree derived from the alignment. Empty and full circles indicate events of duplication of the small (‘GP’) and large (‘QPGFNGV’) pSTRs, respectively. (**C**) AlphaFold model of the *A. thaliana* PQT3L protein structure (AF-B9DFV2-F1). Labels indicate amino acid positions and names of various sequence annotations given by UniProtKb, and of the two highlighted regions with variable pSTRs (in pink). Protein identifiers indicated in the alignment and tree (before species name) are from NCBI’s Entrez Protein.

## Data Availability

pSTR can be accessed from http://cbdm-01.zdv.uni-mainz.de/~munoz/pSTR (accessed on 5 July 2023), and a standalone version of the code downloaded from a dedicated GitHub repository https://github.com/pmiemun/pSTR.

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
