# Peer review of "Evolutionary Study of Protein Short Tandem Repeats in Protein Families"

_biomolecules, 2023, doi:10.3390/biom13071116_

Round 1

Reviewer 1 Report

Sometimes repeat areas are mis-aligned. It is possible that some repeats are missed when alignments are made in the 3rd step of the pipeline. Could the authors explain whether this phenomenon was an issue and how they addressed it? 

In Figure 3A, there are similar repeat units in blue. Are these detected by the pipeline or are they just highlighted for illustration? How are such variant repeats handled by the pipeline?  How are possible smaller repeat units handled by the pipeline if mutations are allowed in the repeat units? 

Why is the set of orthologs restricted to just 70 families? 

I could not access the web tool. This needs to be made accessible to enable a complete review. 

Author Response

1) Sometimes repeat areas are mis-aligned. It is possible that some repeats are missed when alignments are made in the 3rd step of the pipeline. Could the authors explain whether this phenomenon was an issue and how they addressed it?

While it is true that repeats can be mis-aligned, our pipeline has two aspects that limit them: (1) to detect the pSTR, we do pairwise alignments, which simplify the alignments thus reducing errors, and (2) we use homologous proteins, so that their sequences are relatively (or very) similar. Both elements limit the theoretical mis-alignments of the repeats. We have added this clarification to the revised manuscript in section 3.1.

2) In Figure 3A, there are similar repeat units in blue. Are these detected by the pipeline or are they just highlighted for illustration? How are such variant repeats handled by the pipeline? How are possible smaller repeat units handled by the pipeline if mutations are allowed in the repeat units?

The blue units are highlighted for illustration, as close variants of the units highlighted in pink. However, as the pipeline locates identical repeats, it is also true that it detects both “QPGFNGV” (in XP_006279986-1_Crubella and in NP_199554-2_Athaliana), and “QPGFNGF” (in XP_018471595-1_Rsativus and in ABD65123-1_Boleracea). The pipeline does not allow for mutations in the repeat units. This example can be found in the web tool as an “example dataset”.

3) Why is the set of orthologs restricted to just 70 families?

We used a reference dataset of manually curated protein families called OrthoBench, which is the standard benchmark to assess the accuracy of orthogroup inference methods. We used all protein families provided by this database. We added an explanation in section 3.3 of the manuscript. 

4) I could not access the web tool. This needs to be made accessible to enable a complete review.

We are sorry for this inconvenience. We have checked the three times the tool’s web site is mentioned in the manuscript and it seems to be well written; furthermore, the other reviewers had no issue when accessing it. We invite the reviewer to try again by copying the site’s web direction: http://cbdm-01.zdv.uni-mainz.de/~munoz/pSTR/.

Reviewer 2 Report

The authors of the manuscript 'Evolutionary study of protein Short Tandem Repeats in protein families'
describe the occurrence of tandem repeats in proteins and developed a tool to check proteins
for the presence of these repeats.

The manuscript is well written and development of the tool and conclusions are clear.
The tool can be very helpful for research on proteins involved in phase separation.

Some minor comment
The authors see a difference in used amino acids use in repeats for different species.
How would the difference in codon usage between species affect this observation? Could the authors address this issue?

The use of Alphafold is generally not very good for prediction of structures of disordered or partially disordered proteins.
What we can say is that if Alphafold does not show a well folded protein, the protein is most likely disordered.

Author Response

1) The authors see a difference in used amino acids use in repeats for different species. How would the difference in codon usage between species affect this observation? Could the authors address this issue?

We do not think the codon usage plays a major role in relation with the amino acid usage in these repeats. The background of the amino acid usage was calculated per proteome independently, and thus each background can be compared with the results for the pSTR located in a species. The amino acid usage in the 12 proteomes is also very similar.

2) The use of Alphafold is generally not very good for prediction of structures of disordered or partially disordered proteins. What we can say is that if Alphafold does not show a well folded protein, the protein is most likely disordered.

Exactly; we agree. The AlphaFold prediction is not really important for this case and it is only shown to illustrate that most of the protein and repeated regions are likely disordered. We have added a comment about this in the manuscript, in section 3.3. 

Reviewer 3 Report

In “Evolutionary study of protein Short Tandem Repeats in protein families” by Mier & Andrade-Navarro, the authors examine the conservation patterns of short (less than 10 residues) protein tandem repeats (pSTR) in families of proteins. They look at these pSTR 70 groups of orthologous proteins in 12 bilaterian animal species, ignoring sequences with ambiguous “X” amino acid sequences. They find about 3% of the residues in the proteome contain pSTR, which is possibly an important observation (see below). They also identify propensities for amino acids to appear in pSTR, which is also an important observation, although tempered a bit by the fact that other than lysine, there is a rough correlation with the essential/non-essential amino acid set in animals which likely corresponds at least partially to the author's concept of “evolutionary pressure” (see below). The authors discuss a relationship between pSTR variation and liquid-liquid phase separation Then they use the A. thaliana PQT3L protein as an illustrative example of their analytical methods and report a web-based tool that can be used by others to do similar analyses. Overall the paper is well-written and the research well-composed but some issues arise.

Minor Issues:

  1. The title “Evolutionary study” is probably grammatically incorrect and should be “Evolutionary studies...”.

  1. The main focus of the article is pSTR (protein short tandem repeats). While these are clearly repeats, as well as short and protein, are they legitimately tandem? Tandem should mean something like “arranged in a continuous order”. And while there can be some exceptions made for short insertions of a few amino acids in between the repeats and still legitimately call them „tandem”, this is not the case here. The illustration in Figure 1 shows a number of these pSTR in pink, they are often separated by stretches of non-repeating amino acids that are as long or longer than the repeats themselves, so the term „tandem” does not seem to apply here. However, this is generally a semantic issue so it could easily be fixed by explaining in the text why these are considered to be tandem repeats.

  1. Additionally, the authors introduce the term „evolutionary pressures” without really discussing it. How broadly or narrowly are the authors using the term? A sentence defining what the authors mean by this term should be included.

  2. Supplemental file 1 needs a better explanation in the text or included within the associated zipped folder. Most of the pdf files appear to be simply empty or blank.

  3. Its always hard to make figures with more than about 6 colors, especially for color-blind or older readers. Figure 2 isn't terrible, but it should be improved, It starts with a reasonable light/dark pattern but this doesn't hold up all the way, especially for the Lys/Leu and Trp/Val pairs.

  4. While it is good that the authors chose an non-animal protein for its illustrative example in Figure 3, the low confidence AlphaFold prediction diminishes the impact of the figure. A better example (plant) protein in which AlphaFold is more confident of its abilities would greatly improve the figure.

  5. While I applaud the authors for having a functioning web server for their analysis tool, its also good to have it available as a stand alone downloadable file somewhere as well, perhaps GitHub or on one of the author's laboratory home pages for the future date when the web server also inevitably expires.

Major Issues:

  1. The observation that 3% of animal genomes are pSTR is an interesting and important fact, if it can be supported. And while there is no doubt that the authors do observe these amino acids, the issue that arises is that these are short sequences, so their occurence probability is rather high, as the authors correctly note for the simple 2 residue repeat. And while a 3 residue repeat is less common, its not fantastically rare either., especially if one only sees a single pair of these 3 residue repeats in a protein. There are a number of simple statistical models for determining how often one should encounter a given pattern within a protein of a given length (see for example doi.org/10.3390/biom12060793). This manuscript has the added issue of having to deal with multiple repeat lengths, sequence compositions and protein lengths, making it a rather thorny issue. The authors must make some attempt to address how often such patterns might be expected to occur randomly as compared to how often they are observed. The fact that they are conserved does not exempt this concern either, as its not impossible for a random repeat to occur within highly conserved, functional regions and simply get „grandfathered” into the sequence due to the need to conserve the region.

  2. The authors largely restrict themselves to bilaterian metazoa in their thorough analysis. This raises 2 issues:

    A) The authors talk about variation and conservation of the pSTR in the metazoa. And while the genomes examined are varied, there is little no analysis of whether any evidence of phylogenic lineage plays any role in the variation or conservation that I could discern other than a mention of drosophila having more pSTR than other metazoa and some amino acid frequencies. Can one find variation along the mammal line for a specific set of pSTR? Are most of them conserved among the synapsids and variant in the sauropsids or vice versa? Or are different sets of repeats conserved in both? Does lineage matter at all? If not, the previous issue about the random baseline level of pSTR generation begins to be more concerning....

    B) The authors analyze the propensity for each amino acid to be found in pSTR. However, this propensity is clouded by its similarity to the essential/non-essential amino acid dichotomy in the animals, which is itself an evolutionary pressure. This conflating factor should be addressed, and this could be done so by whole genome pSTR analysis of a few non-animal genomes, especially plant and fungal ones in which diet is not the only source of a number of amino acids.

Author Response

1) The title “Evolutionary study” is probably grammatically incorrect and should be “Evolutionary studies...”.

We disagree with the reviewer. There is actually only one evolutionary study described in the manuscript, but it is divided in two sections: in section 3.2 we look for pSTRs in the different proteomes, and in section 3.3 we compared their pSTRs to describe unit variation.

2) The main focus of the article is pSTR (protein short tandem repeats). While these are clearly repeats, as well as short and protein, are they legitimately tandem? Tandem should mean something like “arranged in a continuous order”. And while there can be some exceptions made for short insertions of a few amino acids in between the repeats and still legitimately call them „tandem”, this is not the case here. The illustration in Figure 1 shows a number of these pSTR in pink, they are often separated by stretches of non-repeating amino acids that are as long or longer than the repeats themselves, so the term „tandem” does not seem to apply here. However, this is generally a semantic issue so it could easily be fixed by explaining in the text why these are considered to be tandem repeats.

The “tandem” in pSTR (protein Short Tandem Repeats) refers to repetitions of the same unit. In Figure 1, for example, there are four different pSTRs, with units “ML”, “ACD”, “PQ” and “IL”. They are not related in any way, they happen to be close in the example of Figure 1 for illustration purposes.

3) Additionally, the authors introduce the term „evolutionary pressures” without really discussing it. How broadly or narrowly are the authors using the term? A sentence defining what the authors mean by this term should be included.

We mean that in a given protein family there may be some selective advantage given by the variation of the tandem repeats. We have added this explanation in the Introduction of the revised manuscript.

4) Supplemental file 1 needs a better explanation in the text or included within the associated zipped folder. Most of the pdf files appear to be simply empty or blank.

We have added a README file to the Suppl.File1 folder to explain the different files each folder may contain. There is also a detailed legend for Suppl.File1 in the manuscript, in section “Supplementary Materials” (following the journal’s guidelines). We have also deleted the empty files related to orthologous groups with no pSTR with unit number variation.

5) Its always hard to make figures with more than about 6 colors, especially for color-blind or older readers. Figure 2 isn't terrible, but it should be improved, It starts with a reasonable light/dark pattern but this doesn't hold up all the way, especially for the Lys/Leu and Trp/Val pairs.

We thank the reviewer for the input; it is really difficult to select 20 colors that are different enough to be unique and distinguishable in a figure. We have changed some of the colors to improve it, and produced a revised Figure 2. In any case, the order of the bars follow the alphabetical order of the amino acids (one letter code), making it easier to identify them.

6) While it is good that the authors chose an non-animal protein for its illustrative example in Figure 3, the low confidence AlphaFold prediction diminishes the impact of the figure. A better example (plant) protein in which AlphaFold is more confident of its abilities would greatly improve the figure.

The AlphaFold prediction is not really important for this case and it is only shown to illustrate that most of the protein and repeated regions are likely disordered. We have added a comment about this in the manuscript, in section 3.3.

7) While I applaud the authors for having a functioning web server for their analysis tool, its also good to have it available as a stand alone downloadable file somewhere as well, perhaps GitHub or on one of the author's laboratory home pages for the future date when the web server also inevitably expires.

We have created a dedicated GitHub repository and uploaded the standalone code of pSTR; it can be accessed from https://github.com/pmiemun/pSTR. We have added this information to the revised manuscript, in section “Data Availability Statement”.

8) The observation that 3% of animal genomes are pSTR is an interesting and important fact, if it can be supported. And while there is no doubt that the authors do observe these amino acids, the issue that arises is that these are short sequences, so their occurence probability is rather high, as the authors correctly note for the simple 2 residue repeat. And while a 3 residue repeat is less common, its not fantastically rare either., especially if one only sees a single pair of these 3 residue repeats in a protein. There are a number of simple statistical models for determining how often one should encounter a given pattern within a protein of a given length (see for example doi.org/10.3390/biom12060793). This manuscript has the added issue of having to deal with multiple repeat lengths, sequence compositions and protein lengths, making it a rather thorny issue. The authors must make some attempt to address how often such patterns might be expected to occur randomly as compared to how often they are observed. The fact that they are conserved does not exempt this concern either, as its not impossible for a random repeat to occur within highly conserved, functional regions and simply get „grandfathered” into the sequence due to the need to conserve the region.

As the reviewer states, it is a very difficult calculation, as there are multiple pSTR with various lengths, compositions, and unit number. We located a total of 1,304,253 pSTRs in the 12 proteomes, and 75,402 different units. They may also overlap (i.e. sequence “WACDEACDEAM” includes two pSTRs, one with unit “ACDE” repeated twice, and another one with unit “CDEA” also repeated twice). For our calculation, what we did was take into account the percentage of residues in each proteome that participate in at least one pSTR, 9/11 residues in the previous example.
But we do not think this calculation is flawed. For the sake of the argument, let’s take the most prevalent amino acid in human, leucine, with a 0.1 amino acid usage. The shortest pSTR considered would be of length 2, and with two units: sequence “LLLL”. Considering the chance of finding each amino acid independently, the probability of finding it would be 0.1*4 = 0.01%. In any case, we believe repeating this calculation with the 75,402 different units would not solve the problem of the overlapping of the pSTRs.
We have added a clarification about this calculation in section 3.2 of the revised manuscript.

9) The authors largely restrict themselves to bilaterian metazoa in their thorough analysis. This raises 2 issues:

A) The authors talk about variation and conservation of the pSTR in the metazoa. And while the genomes examined are varied, there is little no analysis of whether any evidence of phylogenic lineage plays any role in the variation or conservation that I could discern other than a mention of drosophila having more pSTR than other metazoa and some amino acid frequencies. Can one find variation along the mammal line for a specific set of pSTR? Are most of them conserved among the synapsids and variant in the sauropsids or vice versa? Or are different sets of repeats conserved in both? Does lineage matter at all? If not, the previous issue about the random baseline level of pSTR generation begins to be more concerning.…

We understand that there is a lot of further studies needed to assess the association of particular rates of pSTR evolution and various taxa, but in this work, we wanted to point to the importance of pSTR on the length variation of IDRs, indicate that they could be related to functional adaptability, and provide a general and simple method to study pSTR evolution in families.  

B) The authors analyze the propensity for each amino acid to be found in pSTR. However, this propensity is clouded by its similarity to the essential/non-essential amino acid dichotomy in the animals, which is itself an evolutionary pressure. This conflating factor should be addressed, and this could be done so by whole genome pSTR analysis of a few non-animal genomes, especially plant and fungal ones in which diet is not the only source of a number of amino acids.

While this analysis could be interesting, we think it would be out of scope since we wanted to be coherent in this work with the species featured in OrthoBench. We have added these points to the discussion.

Round 2

Reviewer 1 Report

The comments have been addressed. 

The server is working and useful. 

Author Response

We thank again the reviewer for his/her comments.

Author Response

We thank again the reviewer for his/her insightful comments.

*) This is an important definition and a needed clarification for this manuscript, please include text to this effect in the manuscript.

We have included the clarification in the manuscript.